# Parental Socialization and Development of Chinese Youths: A Multivariate and Comparative Approach

**DOI:** 10.3390/ijerph16101730

**Published:** 2019-05-16

**Authors:** Jerf W. K. Yeung, Eileen Y. H. Tsang, Hui-Fang Chen

**Affiliations:** Department of Social and Behavioural Sciences, City University of Hong Kong, Kowloon, Hong Kong, China; eileen@cityu.edu.hk (E.Y.H.T.); hfchen@cityu.edu.hk (H.-F.C.)

**Keywords:** authoritative parenting, positive family processes, multi-informant approach, youth development

## Abstract

Parental socialization has been recently reported as a multifaceted concept, which includes parenting practices and family processes. Nevertheless, prior family research generally treated parental socialization tantamount to parenting behavior only and overlooked its different effects on multiple youth outcomes simultaneously, especially in the Chinese population. This study, with a sample of 223 Chinese parent-youth dyads (80.7% mothers; 55.6% male youths; mean_age_ = 16.7 years), found that both authoritative parenting and positive family processes, as measured by a multi-informant approach, significantly predicted higher self-esteem, self-control, future orientation, other perspective taking and lower externalizing problem behavior of Chinese youths concomitantly. Furthermore, youth self-esteem was found to significantly mediate the effects of authoritative parenting and positive family processes on their self-control, future orientation, other perspective taking and externalizing problem behavior, and different facets of parental socialization significantly predicted the youth outcomes differentially. Results of this study highlight importance of considering the multifaceted nature of parental socialization and interrelations of youth development.

## 1. Introduction

Theoretically and practically, family experiences have profound influences on youth development, which are mainly through the process of parental socialization [1,2,3,4]. Nevertheless, prior family studies generally considered parenting practices as tantamount to parental socialization, ignoring its multifaceted nature [5,6]. Recent pertinent research pointed out that parental socialization is a multi-dimensional construct, including both parenting practices and family processes [7,8,9,10]. On the other hand, although empirical research supported beneficial effects of authoritative parenting and positive family processes on youth development [6,7,10], some researchers cast skepticism on their salutary contributions to Chinese youths in parallel as more hierarchical parent-child relationships and disciplinarian parental socialization prevailed in Chinese culture [4,11]. Furthermore, youth development involves multiple aspects of cognitive, psychological and behavioral changes synchronously [10,12]. However, extant research predominantly examined only one or two youth outcomes in a single study [1,2,5], which hence tends to overlook diverse and interrelated aspects of youth development.

In all, this study intended to investigate effects of both authoritative parenting and positive family processes on developmental outcomes of self-esteem, self-control, future orientation, other perspective taking and externalizing problem behavior among Chinese youths, in which youth self-esteem is expected to mediate the effects of parental socialization on the youth outcomes and authoritative parenting and positive family processes are thought to have different effects on different youth outcomes.

## 2. Parental Socialization and Development of Chinese Youths

Parents are the main and most intimate socialization agent fundamentally influential of youths’ various aspects of development, in which actions of parental figures performed and the socialization environments they created are generally referred as parental socialization [2,7,9]. Nevertheless, prior family research tended to treat parenting practices of parental caregivers tantamount to parental socialization, therefore overlooking the effects of other parental socialization facets on youth development [1,7,9]. Consonantly, recent research revealed that parental socialization is a multifaceted concept comprising of not only parenting behaviors but also family processes. Manifestly, parenting practices refer to the ways, instructions and rearing methods that parents adopted to socialize their offspring, which are hence more directive and hierarchical embedded in parent-child interactions [10,13]. However, family processes mean the general relational quality, communication patterns and home climate that parents cultivated and established within the family realm as an overall socialization atmosphere, which is thus more reciprocal and interdependent in terms of parent-child relationships [8,9]. In sum, both parenting practices and family processes have been empirically proved to affect youth development profoundly, for which authoritative parenting and positive family processes are two parental socialization facets beneficially contributing to youth development [2,6,10,13].

Specifically, authoritative parenting, as demonstrating both high parental demandingness and responsiveness concurrently, has been corroborated to positively contribute to youths’ establishment a self-identity of worth, competence and respect, development of cognitive and psychological strengths, and decrease of mental health problems and behavioral maladjustment [1,4,6,12]. Conceptually, social learning theory and social control theory can help explain the beneficial effects of authoritative parenting on youth development, in which youths’ self-concept, value orientations, cognitive approaches, psychological traits and behavioral choices are cultivated in the processes of social learning and controlling and parents are the prime socialization agent to foster and steer development of their youth children by instructions, rules, directions, support and expectations set in their parenting practices [1,14,15]. Empirically, Hirata and Kamakura (2018) found that authoritative parenting was significantly related to higher personal growth initiative and self-esteem among university students and such relationships did not exist in authoritarian and permissive parenting styles. Furthermore, in his meta-analysis, Pinquart [6] reported that authoritative parenting had substantial negative effects on reducing aggressive and delinquent behaviors of children and adolescents. Similarly, in their systematic review, Rose, Roman, Mwaba and Ismail [16] concluded that authoritative parenting was associated with fewer internalizing symptoms among diverse child samples. In addition, a recent study by Yeung and colleagues [4] found that authoritative parenting significantly predicted less externalizing and internalizing problems and higher self-regulatory and perspective taking behaviors of Chinese youths. Hence, it is anticipated that authoritative parenting would beneficially contribute to youths’ various developmental outcomes.

Moreover, albeit limited in number, certain recent empirical studies supported beneficial impacts of positive family processes on youth development [3,8,9]. In this study, positive family processes indicate intimate relationships, efficient communications, mutual support and harmonious climate existing in the home environment that are favorable for healthy youth development [7,10]. Pertinently, the earliest work conducted by Brody, Stoneman and Flor [16] showed that effective family processes positively contributed to youths’ development of self-regulation that in turn predicted their better academic competence and lower internalizing and externalizing problems. In another study, Yabiku, Axinn and Thornton [17] found that positive family processes in terms of family cohesion, parent-child closeness and parental integration prospectively and significantly predicted higher youth self-esteem, which sustained even in their adulthood. Moreover, based on a large sample of 8997 youths from Hungary (*n* = 826), Japan (*n* = 344), Netherlands (*n* = 1244), Switzerland (*n* = 3819) and the United States (*n* = 2764), Vazsonyi and Bellisiton [3] found that positive family processes significantly predicted higher self-control and lower delinquency across the youth subsamples, vindicating external validity of positive family processes in relation to youth development. Recently, Yeung et al. [10] also reported that positive family processes had significant effects on reducing both internalizing and externalizing symptoms of Chinese youths. Relevantly, observational learning theory posits that cognitive, psychological and behavioral development of youths hinge on their most direct and intimate family experiences, interactions and climate established and cultivated in the process of parental socialization [10,18], in which positive family processes render youths good examples of self-restraint, respectfulness, problem-solving, resilience, hope and mutual support to be modeled and learnt by observations and molding. As authoritative parenting and family processes are two different facets of parental socialization, research should investigate their respective effects on youth development. Hence, in this study it is expected that positive family processes would beneficially predict youths’ various developmental outcomes.

### 2.1. Synchronicity of Youth Development and the Mediation of Self-Esteem

Undeniably, adolescence is a transformative period for youths to undergo profound cognitive, psychological and behavioral development, in which various aspects of youth outcomes are happened in synchronicity and occurred in interrelations [8,12]. Nevertheless, prior research has seldom looked into how parental socialization simultaneously shapes multiple developmental outcomes of youths. In fact, various youth outcomes are found to correlate with each other. In their latest empirical study, Yeung et al. [4] found that self-concept, perspective taking behavior, self-control, internalizing and externalizing problems of Chinese youths were significantly interrelated together. In addition, a study by Shi et al. [11] found significant correlations between self-esteem, loneliness and internet addiction of Chinese youths. Hence, as multiple youth outcomes are coexisted and interrelated with each other, it is plausible to examine the effects of parental socialization on multiple youth outcomes simultaneously in a single study. In this study, both authoritative parenting and positive family processes are anticipated to predict self-esteem, self-control, future orientation, perspective taking behavior and externalizing problems of Chinese youths, in which significant interrelations of the youth outcomes are expected.

In addition, youths’ self-esteem is believed to mediate the effects of authoritative parenting and positive family processes on their self-control, future orientation, perspective taking behavior and externalizing problem behavior. Generally, self-esteem is an important cognitive and psychosocial construct that may sway how individuals evaluate themselves, judge others and interpret external social events around them, which coheres with the perspective of self-referent cognitions suggesting that youths’ emotional expressions, psychological responses, behavioral reactions, apprehension of external environment and future aspiration are all essentially contingent on their referring to self-identity, worth and competence [19]. Albeit other youths’ cognitive and psychological traits, such as youth self-control, hold the possibility to mediate the relationships between parental socialization and youth development, youth self-esteem is considered as a crucial mediator, which is because self-image and concept are found to be more stable and influential on other cognitive, psychosocial and behavioral choices and responses, leading to an individual’s overall development and status [11,17]. Thus, if youths are of higher self-esteem, they would develop better self-control, future orientation, perspective taking behavior and lower externalizing problems [20,21]. Furthermore, as family is a fundamental nurturing context to develop youth self-esteem mainly through parental socialization [1,17]; it is believed that youth self-esteem would mediate the relationships between parental socialization and youth outcomes. Empirically, Li and Wang [22] found that self-esteem significantly mediated the effects of authoritative parenting and family relation on Chinese youths’ development of leadership. Moreover, Shi et al. [11] corroborated that self-esteem was a significant mediator linking the associations between effective family functioning and youths’ loneliness and internet addiction. In sum, youth self-esteem is expected to mediate the effects of authoritative parenting and positive family processes on their self-control, future orientation, perspective taking behavior and externalizing problems.

### 2.2. Comparing the Effects of Authoritative Parenting and Positive Family Processes

In this study, parental socialization is seen as a multifaceted concept consisting of both authoritative parenting and positive family processes, which are expected to have different effects on youth development due to their distinctive socialization nature and characteristics [2,6,8]. As aforementioned, parenting practices are more directive and instructional, which manifestly pivot on rules, standards and norms set by parents to steer development of their offspring; and family processes, on the other hand, are more reciprocal and mutual, which conspicuously rest on the supportive relationship, communication efficiency and interaction quality established between parents and their children. Therefore, it is plausible that these two facets of parental socialization may contribute to youth development differentially. Specifically, youth self-esteem is a cognitive and psychosocial construct that is apparently cultivated and shaped through constant social experiences, interpersonal interactions and establishment of human trust and worth within the family realm [1,20]. Moreover, family relationships and home climate provide one of the most robust modeling environments influential on youths’ behavioral patterns and choices [2,7,9], especially for out-of-home behavioral decisions and manners, which need individual sagacity and judgment to assess and select correct responses [3,7], e.g., deciding to engage in deviance or not. Thereby, incessant and prevailing experiences of family processes are expected to be more influential on youths’ self-esteem and externalizing problem behavior than that of authoritative parenting which occurs more conditionally and situationally. In addition, self-control, future orientation and other perspective taking of youths need to be learnt and acquired or even molded through parental direct discipline, instructions, reminders and suggestions. This is because self-control involves self-regulatory skills and capability, future orientation is related to planful actions and long-term goals, and other perspective taking entails considerateness and solicitousness toward others, which all necessitate direct coaching, inculcation, molding and support of parenting practices [5,16,23]. Hence, authoritative parenting is expected to be more strongly contributive to youth self-control, future orientation and other perspective taking behavior than that of positive family processes.

## 3. The Current Study

On the whole, this study considers parental socialization as a multifaceted construct consisting of both parenting practices and family processes, which would significantly contribute to self-esteem, self-control, future orientation, perspective taking behavior and externalizing problem behavior among Chinese youths simultaneously. Moreover, youth self-esteem is believed to mediate the effects of parental socialization on their developmental outcomes. Furthermore, it is expected that authoritative parenting and positive family processes would shape the youth outcomes differentially, in which the former has stronger effects on youth self-control, future orientation and other perspective taking behavior, and the latter has stronger effects on youth self-esteem and externalizing problem behavior. Taken together, this study has the following hypotheses:

**Hypothesis** **1.**
*Authoritative parenting would contribute to higher self-esteem, self-control, future orientation and other perspective taking and lower externalizing problem behavior of Chinese youths.*


**Hypothesis** **2.**
*Positive family processes would contribute to higher self-esteem, self-control, future orientation and other perspective taking and lower externalizing problem behavior of Chinese youths.*


**Hypothesis** **3.**
*Self-esteem of Chinese youths would mediate the effects of authoritative parenting and positive family processes on their self-control, future orientation, other perspective taking and externalizing problem behavior.*


**Hypothesis** **4.**
*Comparatively authoritative parenting would have stronger effects on self-control, future orientation and other perspective taking of Chinese youths than that of positive family processes, and, in contrast, positive family processes would have stronger effects on self-esteem and externalizing problem behavior of Chinese youths than that of authoritative parenting.*


In testing the above-mentioned hypotheses, certain pertinent and confounding sociodemographic covariates are controlled, which include family composition, family welfare dependence, number of family members, youth gender, age, current status representing in study or work and educational levels. Research showed that youths from single-parent and welfare-dependent families as well as those with more siblings and family members are in more unfavorable conditions for positive development [3,10,24]. Additionally, empirical literature pointed out that female youths tend to have higher psychological difficulties and their male counterparts are apt to have more behavioral problems but higher self-esteem [2,16]. Moreover, youths of older ages, being in student status rather than employment and of higher educational attainment would perform better psychologically and behaviorally than their younger aged, working and lower educated counterparts [3,4,7,17]. Therefore, all these sociodemographic covariates are adjusted in the analysis.

## 4. Methods

### 4.1. Sample and Data

This study employed data obtained from a community sample of 223 Chinese parent-youth dyads. The sample was collected with the help of 43 local churches located in three main geographic locations of Hong Kong. The selection criteria of the parent-youth dyads included that the parents must have been living with the youth participants consecutively for at least five years and as the main caregivers in the family. Moreover, the parent participants needed to be the biological mothers or fathers of the youth participants. For youth participants, they were required to be within the age range between 14 to 21 years old, connoting in a developmental period of pronounced psychological and behavioral concerns [2,12,16]. For increasing diversity of the Chinese sample, if a targeted family had two or more eligible youth children suitable for the study, the one who had just passed his or her birthday was selected. Nevertheless, if a family had twins of youth children both eligible for the study, the older one was selected to increase sampling variance and randomization. In data collection, two versions of questionnaires, one for parents and one for youth participants, enclosed in a survey package were distributed to the eligible families that showed interest in the study. For confirming individual privacy, the parent-version and youth-version questionnaires were contained in an A5-size envelope with identifiers within the survey package respectively, and the parent-youth dyads were required to insert their completed questionnaires in the A5-size envelopes separately and then returned back to the researchers of this study. At beginning, there were 284 parent-youth dyads who gave consent to participate in the study, and in the end, a sample of 223 parent-youth dyads who gave complete data was used for analysis. The study was approved by the ethical review committee of The Hong Kong Polytechnic University.

### 4.2. Measures

Authoritative parenting was measured by the 10-item Authoritative Parenting subscale of the Parental Authority Questionnaire (PAQ) [25]. Recent studies showed good psychometrical properties and internal reliability [13,23]. An example item is “I will listen to what my children say but will not choose to do something just because they say so”. In this study, a multi-informant approach was used to measure authoritative parenting by combining the scores of parent and youth participants, which is proved to be more liable and can reduce method variance bias [3,10]. Hence, the item was rephrased to “My mother will listen to what the children say but will not choose to do something just because the children say so”. Rephrasing items of existing measures to cater for specific research needs has been common in conducting empirical research [5,10,16]. The reliability alphas for parent and youth participants were excellent in this study, both were α = 0.89.

Positive family processes were measured by the 26-item Family Functioning Style Scale (FFSS) [26], which has been commonly used to tap into effective family communications, support and relationship quality within the family realm [9,27]. Example items are “Our family sticks together no matter how difficult things get” and “We are always willing to ‘pitch in and help each other”. In this study, a multi-informant approach was also used to measure positive family processes by combining the scores of parent and youth participants. Cronbach alphas for parent and youth participants were excellent, both were α = 0.95.

Youth self-esteem was measured by the 6-item Positive Self-Image Scale [28], which was used to measure positive self-concept from a representative sample of youths of diverse ethnic origins [28]. Example items include “You like yourself just the way you are” and “You are doing everything just about right”. Internal consistency was highly satisfactory in this study, α = 0.86.

Youth self-control was measured by the 7-item Good Self-Control Scale from Wills and colleagues [29], which is mainly used to measure youths’ self-regulatory and persistent behavior [29]. Example items include “When I promise to do something, you can count on me to do it” and “I usually think before I act”. In this study, alpha reliability was good, α = 0.75.

Youth future orientation was measured by the 8-item Future Outlook Inventory [30], which is to measure future expectations and aspiration [30]. Example items include “I think about how things might be in the future” and “I make lists of things to do”. Internal reliability in this study was highly satisfactory, α = 0.78.

Youth other perspective taking behavior was measured by the 7-item Consideration of Others Scale [31], which is used to measure youths’ capability of being considerate and understanding others’ points of view [31]. An example item is “I make sure that doing what I want will not cause problems for other people”. Excellent Cronbach alpha was obtained in this study, α = 0.86.

Youth externalizing problem behavior was measured by the 15-item Externalizing Problem Symptoms Scale (EPSS) developed by Yeung [9], which aims to measure deviant acts and externalizing symptoms of Chinese youths and has satisfactory internal consistency [9,10]. Example items include “destroy public property” and “steal things from places other than home”. In this study, Cronbach alpha was good, α = 0.77.

Sociodemographic covariates controlled in this study include family composition, family welfare dependence, number of family members, youth gender, age, current status and educational levels, in which family composition (2 = non-intact family, 1 = otherwise), family welfare dependence (2 = dependence, 1 = otherwise), youth gender (2 = females, 1 = males) and current status (2 = working, 1 = studying) are dummy variables, and number of family members, youth age and educational levels are count variables.

### 4.3. Analytic Procedures

As the youth outcomes of self-esteem, self-control, future orientation, other perspective taking behavior and externalizing problem behavior are expected to be interrelated, multivariate linear regression is suited for analyzing the data, which can take the advantages of accounting for non-independence [32]. In conducting multivariate linear regression, all the youth outcomes are concurrently regressed on authoritative parenting and positive family processes and pertinent sociodemographic covariates, in which the form presents:
yik=β0+∑j=1pβjkxij+εik,
where yik∈ℝ is the *k^th^* real valued response for *i^th^* observation, β0∈ℝ is the regression intercept for the *k^th^* response, βjk∈ℝ is the *j^th^* predictor’s regression slope for *k^th^* response, and
xij∈ℝ is the *j^th^* predictor for the *i^th^* observation. As such, error variance expresses as (εi1,…,εim)∼N(0m,Σ) to represent a multivariate Gaussian error vector. Therefore, the regression model is multivariate because m > 1 response variables and *p* > 1 predictor variables. In addition, t-statistic is used to compare different effects of authoritative parenting and positive family processes on the youth outcomes by setting equality of parameters, which is t=β1−β2S(β1−β2)
where β_1_ and β_2_ indicate the regression estimates of the β_1_ and β_2_ slopes set for comparison, that is regression coefficients of authoritative parenting and positive family processes, and S(β1−β2) represents the estimated difference in standard errors of the regression slopes of β_1_ and β_2_.

## 5. Results

Table 1 shows correlations of the study variables, in which both authoritative parenting and positive family processes were significantly correlated with youth self-esteem, self-control, future orientation, other perspective taking and externalizing problem behavior. Specifically, authoritative parenting was most strongly correlated with youth other perspective taking, r = 0.346, *p* < 0.001, and least strongly related to youth externalizing problem behavior, r = −0.236, *p* < 0.001, and the correlations between authoritative parenting and youth self-esteem, self-control and future orientation were from r = 0.237 to 0.330, *p* < 0.001. In addition, positive family processes were most strongly correlated with youth self-esteem, r = 0.349, *p* < 0.001, and least strongly correlated with youth externalizing problem behavior, r = −0.251, *p* < 0.001, and the correlations between positive family processes and youth self-control, future orientation and other perspective taking were from r = 0.266 to 0.303, *p* < 0.001. Moreover, self-esteem, self-control, future orientation, other perspective taking and externalizing problem behavior of Chinese youths were all significantly correlated with each other from r = −0.171 to 0.600, *p* < 0.001, which support the use of multivariate linear regression modeling to analyze the study relationships. Notably, authoritative parenting was strongly correlated with positive family processes, r = 0.719, *p* < 0.001, which, however, is within the acceptable correlational threshold, that is r ≤ 0.80, for preclusion of concern of collinearity. For further proving that authoritative parenting and positive family processes are two discrete but mutually related family constructs, both confirmatory factor analysis with item parcels and multiple indicators multiple causes (MIMIC) modeling were applied. First, due to the scales used to measure authoritative parenting and positive family processes having many items, Authoritative Parenting subscale of the Parental Authority Questionnaire was randomly parceled into three indicators starting by a seed number of 3, and Family Functioning Style Scale was parceled into four indicators by a seed number of 4, in which the latent factors of authoritative parenting and positive family processes were allowed to be correlated and MIMIC modeling permitted regressing covariates of family composition, family welfare dependence, number of family members, youth gender, age, current status and educational levels on the factor model. As a result, the confirmatory factor analysis model obtained an excellent model-data fit, in which CFI (Comparative Fit Index) = 0.998, TLI (Tucker Lewis index) = 0.997, RMSEA (Root Mean Square Error of Approximation) = 0.028, X^2^ = 15.309, df = 13, *p* = 0.288, with factor loadings ranged from λ = 0.829 to 0.895 for authoritative parenting and factor loadings ranged from λ = 0.884 to 0.923 for positive family processes. In addition, the MIMIC model also obtained a very good model-data fit, CFI = 0.994, TLI = 0.991, RMSEA = 0.029, X^2^ = 57.101, df = 48, *p* = 0.172, with factor loadings ranged from λ = 0.826 to 0.900 for authoritative parenting and factor loadings ranged from λ = 0.885 to 0.921 for positive family processes.

In Table 2, multivariate linear regression was applied to predict the effects of authoritative parenting on youth outcomes in a single model while controlling for pertinent sociodemographic covariates, in which authoritative parenting significantly predicted Chinese youths’ higher self-esteem, β = 0.258; self-control, β = 0.312; future orientation, β = 0.273 and other perspective taking, β = 0.346, *p* < 0.001, and lower externalizing problem behavior, β = −0.206, *p* < 0.01. The whole model is significantly different from zero, model X^2^ = 377.729, *p* < 0.001, and explained 11.1%, 14.4%, 17%, 15.9% and 18.3% of variances of youth self-esteem, self-control, future orientation, other perspective taking and externalizing problem behavior, respectively. In addition, some sociodemographic covariates of Chinese youths had significant effects on their developmental outcomes, which include the effects of family welfare dependence and youth educational levels on youth self-control, β = 0.142 and 0.244, *p* < 0.05, and the effects of youth age and educational levels on future orientation, β = −0.236 and 0.335, *p* < 0.05 and 0.01, as well as the effects of youth gender and current status on their externalizing problem behavior, β = −0.196 and 0.274, *p* < 0.01 and 0.05.

Multivariate linear regression modeling was also used to investigate the effects of positive family processes on youth outcomes (Table 3), in which positive family processes significantly predicted Chinese youths’ higher self-esteem, β = 0.326; self-control, β = 0.282; future orientation, β = 0.220; other perspective taking, β = 0.279 and lower externalizing problem behavior, β = −0.233, *p* < 0.001. The whole model is significantly different from zero, model X^2^ = 377.926, *p* < 0.001, and the model explained 15.1%, 13.1%, 14.8%, 12.5 and 19.6% variances of youth self-esteem, self-control, future orientation, other perspective taking and externalizing problem behavior, respectively. For sociodemographic covariates, female youths had significantly higher future orientation, β = 0.131, *p* < 0.05, and lower externalizing problem behavior, β = −0.211, *p* < 0.01. However, older youths had lower future orientation, β = −0.287, *p* < 0.05; and youths of working status had more externalizing problem behavior, β = 0.265, *p* < 0.05. Furthermore, youths of higher educational levels showed more self-control, β = 0.275, *p* < 0.05, future orientation, β = 0.368, *p* < 0.01 and other perspective taking, β = 0.249, *p* < 0.05.

In Table 4, youth self-esteem was entered as mediator for the relationships between authoritative parenting and youth outcomes. Results showed that youth self-esteem significantly predicted their self-control, β = 0.428; future orientation, β = 0.251; other perspective taking, β = 0.348, *p* < 0.001 and externalizing problem behavior, β = −0.170, *p* < 0.01. Nevertheless, authoritative parenting still significantly contributed to higher self-control, β = 0.202, and future orientation, β = 0.208, *p* < 0.01, and other perspective taking, β = 0.256, *p* < 0.001, as well as lower externalizing problem behavior of Chinese youths, β = −0.162, *p* < 0.05. Manifestly, after incorporation of youth self-esteem as mediator, explained variances increased to 30.7% for youth self-control, 22.6% for future orientation, 26.6% for other perspective taking and 20.9% for externalizing problem behavior. Results of indirect-effect analysis showed that youth self-esteem significantly mediated the associations between authoritative parenting and youth self-control, β_ind_ = 0.110, *p* < 0.01; future orientation, β_ind_ = 0.065, *p* < 0.05; other perspective taking, β_ind_= 0.090, *p* < 0.01 and externalizing problem behavior, β_ind_ = −0.044, *p* < 0.05 (Table 5). Nevertheless, effect sizes indicate that youth self-esteem only had large indirect effect on self-control, k^2^ = 0.190, and medium indirect effects were found for future orientation, k^2^ = 0.146, other perspective taking, k^2^ = 0.151 and externalizing problem behavior, k^2^= −0.086.

Table 6 presents the full model of youth self-esteem and positive family processes in prediction of youth outcomes. Apparently, youth self-esteem significantly predicted their higher self-control, β = 0.432; future orientation, β = 0.259; other perspective taking, β = 0.358, *p* < 0.001 and lower externalizing problem behavior, β = −0.150, *p* < 0.05. However, significant effects of positive family processes on youth outcomes still held, which are β = 0.142 for youth self-control and β = 0.135 for future orientation, *p* < 0.05, as well as β = 0.163 for other perspective taking and β = 0.184 for externalizing problem behavior, *p* < 0.01. After including youth self-esteem as a mediator in the model of positive family processes, explained variances increased to 29% for youth self-control, 20.5% for future orientation, 23.3% for other perspective taking and 21.5% for externalizing problem behavior. Specifically, indirect-effect analysis showed that youth self-esteem significantly mediated the effects of positive family processes on youth self-control, β_ind_ = 0.179, *p* < 0.001; future orientation, β_ind_ = 0.082, *p* < 0.01; other perspective taking, β_ind_ = 0.151, *p* < 0.01 and externalizing problem behavior, β_ind_ = −0.055, *p* < 0.05 (Table 5). Furthermore, effect sizes reveal large indirect effects of youth self-esteem on youth self-control, k^2^ = 0.242, future orientation, k^2^ = 0.191 and other perspective taking, k^2^= 0.197, and medium effect on their externalizing problem behavior, k^2^ = −0.096.

In comparing whether authoritative parenting and positive family processes have different effects on different youth outcomes, *t*-tests found that positive family processes had significantly larger effects on youth self-esteem, β_diff_ = 0.068, and externalizing problem behavior, β_diff_ = −0.027, *p* < 0.001, than that of authoritative parenting (Table 7). On other hand, authoritative parenting had significantly larger effects on youth self-control, β_diff_ = −0.030, future orientation, β_diff_ = −0.053 and other perspective taking, β_diff_ = −0.067, *p* < 0.001, than that of positive family processes. In sum, although both authoritative parenting practices and positive family processes were found to be significantly predictive of various aspects of youth development, their different effects on different youth outcomes were verified. 

## 6. Discussion

Generally, this study found that both authoritative parenting and positive family processes significantly had profound effects on various aspects of youth development, corroborating that different dimensions of parental socialization are all important to overall youth development. Thereby, when examining the effects of parental socialization on youth outcomes, researchers should not only scrutinize the effects of parenting practices, but also consider other parental socialization experiences of youths, e.g., family processes [7,9,31]. In fact, family is the most intimate, nurturing context intensely contributive to youth development of different cultural and ethnic backgrounds, which is less replaceable by other socialization experiences gained from outside, such as school environment and peers’ network [3,17,24]. Relevantly, recent research has supported the universal effects of parental socialization on youth development across countries and cultures [3,8,12]. Hence, it is plausible to find beneficial effects of both authoritative parenting and positive family processes on increasing self-esteem, self-control, future orientation, other perspective taking and reducing externalizing problem behavior among Chinese youths in this study. As a result, parental socialization should not only emphasize on parenting practices solely but also consider other important facts of parental socialization in contribution to positive youth development, such as positive family processes.

Furthermore, this study also confirmed that the effects of parental socialization on multiple youth outcomes are interrelated with each other and happened synchronously. The interrelation of youth outcomes being simultaneously affected by parental socialization gives insights to researchers, educators, social workers and policy makers that a certain aspect of youth maladjustment may reflect their wide-ranging developmental needs [11]. Therefore, a whole-person development approach should be introduced in education, human service and policy systems to let youths be nurtured eclectically and comprehensively, and it should avoid only emphasizing one aspect of youth development while overlooking the importance of other youth outcomes [12,20,24].

In addition, the significant mediation of youth self-esteem in the relationships of parental socialization and youth development explicates youths’ cognitive and psychosocial development bears crucial spill-over effects on their other aspects of development. This resonates with recent empirical studies pointing to the importance of youth self-esteem in swaying youths’ psychological conditions, emotional responses and behavioral choices. Relevantly, Li and Wang [22] found that positive youth self-esteem mediated the effects of authoritative parenting and family relationships on their leadership capability. Moreover, Özdemir et al. [21] reported that self-esteem of Turkish adolescents was a significant mediator between relationships of parental closeness and monitoring and peer approval in relation to their aggression. Further, Shi et al. [11] supported that self-esteem of Chinese youths significantly mediated the effects of family functioning on their internet addiction behaviors. Hence, having youths cultivated a positive self in terms of worth, competence and respect can contribute to multiple aspects of their healthy development. As such, ensuring appropriate parental socialization to enhance positive cultivation of youths’ self-esteem is a fundamental socialization duty belonging not only to parents, but also to other pertinent parties, such as educators and policy makers.

Additionally, this study found that authoritative parenting and positive family processes had different effects on different youth outcomes, in which the former was more strongly related to youth self-control, future orientation, other perspective taking, and the latter was more strongly predictive of youth self-esteem and externalizing problem behavior. As aforementioned, continuing and general family experiences, interactions and relationships would have more persistent influences in shaping youth self-esteem and externalizing problem behavior, because they involve youths’ acquisition of independent interpretations and personal judgements through their daily communications and associations with parents in family processes [4,7,17]. On the other hand, youths’ development of self-regulatory skills, future plans, considerateness and solicitousness needs direct coaching and instructions from parents by parenting practices [5,12,16,23]. Therefore, although both facets of parental socialization shape multiple youth outcomes, their significant effects appear different. Nevertheless, authoritative parenting and positive family processes should be aggregately considered as they are all robustly influential of youth development.

## 7. Conclusions

Conclusively, this study confirms that authoritative parenting and positive family processes, two important facets of parental socialization, are profoundly contributive to various youth outcomes that are also found interrelated in nature. In addition, youth self-esteem is vindicated as an important cognitive and psychosocial construct mediating effects of parental socialization on youth development. Furthermore, in this study different facets of parental socialization are corroborated to have different effects on various aspects of youth development. Nevertheless, this study has several limitations; the Chinese parent-youth dyads recruited in the study is not a representative sample of the Chinese population in Hong Kong, and the cross-sectional design of the research makes causal validity impossible. In addition, data for analysis in this study was obtained from a nonrandom sample of Chinese parent-youth dyads recruited through local churches that may restrict variances of the study variables and representativeness of the results. Furthermore, this study only examined the effects of authoritative parenting and positive family processes on youth outcomes, and the possibility of reciprocal relationships between parental socialization and youth development has not yet explored [33,34]. Future research should consider how parents and youths mutually affect and reinforce each other in a dynamic way to enhance current knowledge on the reciprocity of parental socialization and youth development. In addition, it should be noted that validity of the results would be affected by the measurement instruments and scales adopted to measure the study variables. In this study, authoritative parenting was measured by Authoritative Parenting subscale of the Parental Authority Questionnaire that is more emphasized on measuring parental control rather than a combination of parental demandingness and support. It is suggested to employ more well-established measures in the future to tap into authoritative parenting, such as the Parenting Styles and Dimensions Questionnaire (PSDQ) [35]. In addition, the current study only examined the mediating effects of youth self-esteem on the relationships between parental socialization and youth development, and other possible mediators are yet to be explored. For example, there is a possibility that youth self-control could mediate the effects of parental socialization on youth externalizing behaviors [3,5]. Therefore, future research should consider adopting a multiple-mediation model to incorporate other possible and influential mediators for the associations between parental socialization and youth development. Lastly, parental socialization only referring to authoritative parenting and positive family processes, and other possible facets of family socialization experiences having yet scrutinized in this study, e.g., different parenting styles and parent-youth disclosure, would circumscribe our understandings of how parental socialization affects youth development more comprehensively. If future research can address these limitations, a more dynamic and exhaustive picture of parental socialization and youth development can be attained.

## Figures and Tables

**Table 1 ijerph-16-01730-t001:** Correlations of the study variables.

		1	2	3	4	5	6	7
1	Authoritative parenting	-						
2	Positive family processes	0.719 ***						
3	Youth self-esteem	0.273 ***	0.349 ***					
4	Youth self-control	0.320 ***	0.303 ***	0.501 ***				
5	Youth future orientation	0.330 ***	0.266 ***	0.345 ***	0.600 ***			
6	Youth other perspective taking	0.346 ***	0.295 ***	0.429 ***	0.473 ***	0.485 ***		
7	Youth externalizing problem behavior	−0.236 ***	−0.251 ***	−0.260 ***	−0.329 ***	−0.351 ***	−0.171 ***	-

* *p* < 0.05; ** *p* < 0.01; *** *p* < 0.001.

**Table 2 ijerph-16-01730-t002:** Multivariate linear regression modeling predicting self-esteem, self-control, future orientation, other perspective taking and externalizing problem behaviors of youth by authoritative parenting.

Predictors	Self-Esteem		Self-Control		Future Orientation		Other Perspective Taking		Externalizing Problem Behavior	
	β	95% CI	β	95% CI	β	95% CI	β	95% CI	β	95% CI
Non-intact family	−0.126	−0.274, 0.022	−0.105	−0.251, 0.040	−0.092	−0.234, 0.052	0.003	−0.142, 0.147	0.022	−0.120, 0.164
Welfare dependence	0.087	−0.050, 0.223	0.142*	0.008, 0.276	0.011	−0.121, 0.143	0.065	−0.068, 0.198	−0.052	−0.183, 0.079
Family members	−0.093	−0.234, 0.046	−0.027	−0.165, 0.109	−0.102	−0.237, 0.033	0.046	−0.090, 0.183	0.062	−0.072,0.196
Youth gender	0.019	−0.110, 0.147	0.017	−0.109, 0.143	0.110	−0.014, 0.234	0.025	−0.100, 0.150	−0.196 **	−0.318, −0.072
Youth age	−0.048	−0.286, 0.190	−0.160	−0.394, 0.073	−0.236 *	−0.466, −0.006	−0.072	−0.302, 0.161	0.122	−0.107, 0.349
Youth status	0.115	−0.239, 0.010	−0.072	−0.195, 0.050	−0.052	−0.172, 0.069	−0.103*	−0.224, 0.019	0.274 *	0.154, 0.394
Youth education	0.137	−0.102, 0.376	0.244 *	0.009, 0.478	0.335 **	0.104, 0.566	0.207	−0.026, 0.440	−0.128	−0.357, 0.102
Authoritative parenting	0.258 ***	0.127, 0.389	0.312 ***	0.184, 0.440	0.273 ***	0.147, 0.400	0.346 ***	0.218, 0.473	−0.206 **	−0.331, −0.080
R^2^	0.111		0.144		0.170		0.159		0.183	
Model X^2^ (df)	377.729 (50) ***									

* *p* < 0.05; ** *p* < 0.01; *** *p <* 0.001.

**Table 3 ijerph-16-01730-t003:** Multivariate linear regression modeling predicting self-esteem, self-control, future orientation, other perspective taking and externalizing problem behaviors of youth by positive family processes.

Predictors	Self-Esteem		Self-Control		Future Orientation		Other Perspective Taking		Externalizing Problem Behavior	
	β	95% CI	β	95% CI	β	95% CI	β	95% CI	β	95% CI
Non-intact family	−0.098	−0.245, 0.047	−0.096	−0.243, 0.051	−0.089	−0.243, 0.058	0.007	−0.141, 0.155	0.006	−0.136, 0.148
Welfare dependence	0.070	−0.063, 0.201	0.112	−0.022, 0.245	−0.018	−0.150, 0.115	0.029	−0.105, 0.163	−0.036	−0.165, 0.092
Family members	−0.092	−0.229, 0.044	−0.040	−0.178, 0.097	−0.117	−0.253, 0.020	0.028	−0.110, 0.167	0.065	−0.068, 0.197
Youth gender	0.037	−0.088, 0.162	0.040	−0.086, 0.167	0.131 *	0.006, 0.256	0.051	−0.076, 0.178	−0.211 **	−0.332, −0.088
Youth age	−0.081	−0.311, 0.149	−0.215	−0.448, 0.017	−0.287 *	−0.518, −0.057	−0.136	−0.369, 0.098	0.151	−0.073, 0.375
Youth status	−0.102	−0.223, 0.202	−0.061	−0.184, 0.063	−0.043	−0.165, 0.079	−0.092	−0.215, 0.033	0.265 *	0.146, 0.383
Youth education	0.144	−0.088, 0.376	0.275 *	0.040, 0.510	0.368 **	0.135, 0.600	0.249 *	0.013, 0.484	−0.139	−0.365, 0.087
Positive family processes	0.326 ***	0.201, 0.451	0.282 ***	0.156, 0.409	0.220 ***	0.095, 0.345	0.279 ***	0.153, 0.407	−0.233 ***	−0.355, −0.111
R^2^	0.151		0.131		0.148		0.125		0.196	
Model X^2^ (df)	377.926 (50) ***									

* *p* < 0.05; ** *p* < 0.01; *** *p <* 0.001.

**Table 4 ijerph-16-01730-t004:** Multivariate linear regression modeling predicting self-control, future orientation, other perspective taking and externalizing problem behaviors of youth by their self-esteem and authoritative parenting.

Predictors	Self-Control		Future Orientation		Other Perspective Taking		Externalizing Problem Behavior	
	β	95% CI	β	95% CI	β	95% CI	β	95% CI
Non-intact family	−0.052	−0.183, 0.080	−0.060	−0.199, 0.080	0.047	−0.089, 0.182	0.001	−0.140, 0.141
Welfare dependence	0.105 *	−0.016, 0.226	−0.011	−0.138, 0.117	0.035	−0.089, 0.160	−0.037	−0.167, 0.092
Family members	0.102	−0.112, 0.136	−0.079	−0.210, 0.053	0.079	−0.049, 0.207	0.046	−0.087, 0.179
Youth gender	0.009	−0.105, 0.122	0.105	−0.015, 0.225	0.018	−0.098, 0.135	−0.193 **	−0.313, −0.071
Youth age	−0.140	−0.350, 0.070	−0.224 *	−0.445, −0.001	−0.055	−0.270, 0.162	0.114	−0.112, 0.337
Youth status	−0.023	−0.134, 0.088	−0.023	−0.140, 0.094	−0.063	−0.177, 0.052	0.255 ***	0.136, 0.373
Youth education	0.185	−0.027, 0.397	0.300 *	0.076, 0.524	0.160	−0.059, 0.377	−0.105	−0.330, 0.122
Self-esteem	0.428 ***	0.312, 0.544	0.251 ***	0.129, 0.374	0.348 ***	0.228, 0.467	−0.170 **	−0.294, −0.046
Effective parenting	0.202 **	0.082, 0.321	0.208 **	0.082, 0.335	0.256 ***	0.133, 0.379	−0.162 *	−0.289, −0.034
R^2^	0.307		0.226		0.266		0.209	
Model X^2^ (df)	345.558 (42) ***							

* *p* < 0.05; ** *p* < 0.01; *** *p <* 0.001.

**Table 5 ijerph-16-01730-t005:** Indirect effects of youth self-esteem on the relationships between authoritative parenting and positive family processes in relation to youth self-control, future orientation, other perspective taking and externalizing problem behavior.

**Authoritative Parenting**		**Indirect Effect (** **β_ind_)**	**SE**	**z-Value**	**Effect Size (k^2^)**
1.	Self-control,	0.110 **	0.034	3.246	0.190
2.	Future orientation,	0.065 *	0.025	2.581	0.146
3.	Other perspective taking	0.090 **	0.033	2.700	0.151
4.	Externalizing problem behavior	−0.044 *	0.019	−2.253	0.086
**Positive Family Processes**		**Indirect Effect (β_ind_)**	**SE**	**z-Value**	**Effect Size (** **k^2^)**
1.	Self-control,	0.179 ***	0.045	4.008	0.242
2.	Future orientation,	0.082 **	0.027	3.028	0.191
3.	Other perspective taking	0.151 **	0.047	3.196	0.197
4.	Externalizing problem behavior	−0.055 *	0.026	−2.108	−0.096

* *p* < 0.05; ** *p* < 0.01; *** *p <* 0.001.

**Table 6 ijerph-16-01730-t006:** Multivariate linear regression modeling predicting self-control, future orientation, other perspective taking and externalizing problem behaviors of youth by their self-concept and positive family processes.

Predictors	Self-Control		Future Orientation		Other Perspective Taking		Externalizing Problem Behavior	
	β	95% CI	β	95% CI	β	95% CI	β	95% CI
Non-intact family	−0.053	−0.187, 0.081	−0.063	−0.204, 0.079	0.042	−0.097, 0.181	−0.009	−0.149, 0.132
Welfare dependence	0.082	−0.039, 0.203	−0.036	−0.163, 0.093	0.004	−0.121, 0.130	−0.026	−0.153, 0.101
Family members	0.000	−0.126, 0.125	−0.093	−0.225, 0.040	0.061	−0.068, 0.192	0.051	−0.081, 0.182
Youth gender	0.024	−0.090, 0.139	0.121	0.000, 0.242	0.038	−0.081, 0.157	−0.205 **	−0.325, −0.084
Youth age	−0.180	−0.391, 0.030	−0.267 *	−0.489, −0.043	−0.107	−0.325, 0.112	0.139	−0.083, 0.360
Youth status	−0.017	−0.129, 0.095	−0.017	−0.136, 0.102	−0.055	−0.171, 0.062	0.249 ***	0.131, 0.367
Youth education	0.213 *	−0.001, 0.426	0.330 **	0.105, 0.556	0.197	−0.025, 0.418	−0.117	−0.341, 0.107
Self-esteem	0.432 ***	0.312, 0.552	0.259 ***	0.131, 0.385	0.358 ***	0.233, 0.482	−0.150 *	−0.276, −0.024
Positive family processes	0.142 *	0.021, 0.262	0.135 *	0.008, 0.264	0.163 **	0.038, 0.289	−0.184 **	−0.311, −0.057
R^2^	0.290		0.205		0.233		0.215	
Model X^2^ (df)	336.071 (42) ***							

* *p* < 0.05; ** *p* < 0.01; *** *p <* 0.001.

**Table 7 ijerph-16-01730-t007:** Comparison of effects of authoritative parenting and positive family processes on youth outcomes.

Outcomes		Effects of Positive Family Processes	Effects of Authoritative Parenting	Difference in Betas	*t*-Value
		β (SE)	β (SE)	β_diff_	
1.	Self-esteem	0.258 (0.068) ***	0.326 (0.056) ***	−0.068	−5.667 ***
2.	Self-control	0.312 (0.062) ***	0.282 (0.064) ***	0.030	15.000 ***
3.	Future orientation	0.273 (0.071) ***	0.220 (0.078) ***	0.053	30.000 ***
4.	Other perspective taking	0.346 (0.067) ***	0.279 (0.062) ***	0.067	13.400 ***
5.	Externalizing problems	−0.206 (0.064) **	−0.233 (0.059) ***	−0.027	−5.400 ***

* *p* < 0.05; ** *p* < 0.01; *** *p <* 0.001.

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
