# Peer review of "Parental Socialization and Development of Chinese Youths: A Multivariate and Comparative Approach"

_ijerph, 2019, doi:10.3390/ijerph16101730_

Round 1
Reviewer 1 Report
- dating mistakes.
- little sample.
- poor review.
- Simple data analysis.
- Improved in introduction and discussion
Author Response
After taking comments from reviewer 1, we have revised the current manuscript and have the following amendments:
1. We have revised the theoretical framework in the Introduction part to make the study relationships between parental socialization and youth development in Chinese context more accurate, understandable and convincing
2. Specifically, we have rewritten the mediation of youth self-esteem in the relationships between parental socialization and youth development of self-control, future orientation, other perspective taking and externalizing problem behavior to be more justifiable and testable.
3. Moreover, for comparative purpose we have revised some presentations in the Introduction part to enhance the justifiability of comparting effects of authoritative parenting and positive family processes on different youth outcomes.
4. We have provided more information on the instruments and scales used to measure the study variables, such as youth future orientation.
5. We now have re-analyzed the results by providing 95% CI in the multivariate regression models for enhancing their accuracy.
6. We have also provided new evidence to testify authoritative parenting and positive family processes are two discrete but mutually related family constructs.
7. We have added new information in the Discussion part to address clearly the limitations of the current study.
Reviewer 2 Report
In principle, the topic of the present manuscript is interesting as most studies on associations of parenting styles and family functioning with child outcomes have been done in the US. Therefore, it is interesting to see that similar relations are found in a Chinese sample. Nonetheless, the present manuscript has some problems that need to be fixed before the manuscript can be recommended for publication:
First, I did not find the hypothesis about moderating effects very convincing. In addition, concurrent correlational data do not provide a stringent test for mediation. Many alternative mediation models might fit the data as well. For example, authoritative parenting might promote self-control, which again reduces the level of externalizing problem behavior. In other words, the authors test one of a number of possible mediation models and it is not clear whether the tested model fits the data best.
Second, the statistical test for differences in regression coefficients is not adequate. The authors should provide the 95% confidence intervals of the regression coefficients. If the intervals overlap, this indicates that the associations of authoritative parenting and of positive family relations with the child outcome do not differ significantly. I would guess that this will be the case in most or even all tests.
Third, the authors provide causal interpretations of correlational data which is highly inadequate. Correlations cannot be interpreted as indicating effects or influences of authoritative parenting or positive family relationship on the child outcome as there are also child effects on the quality of family relations and on parenting styles. The authors cite one meta-analysis that has shown that there are bi-directional effects between parenting and parent-child relationship and child outcomes.
Fourth, the authors use a less than optimal measure of authoritative parenting (the Parental Authority Questionnaire). I am aware of the fact that this measure is often used. However, Buri’s scale mainly assesses only control while the conceptualization of authoritative parenting by Maccoby and Martin conceptualize this parenting style by a combination of control and parental warmth. If the authors had used a measure of authoritative parenting that also includes parental warmth, they might have found stronger associations of authoritative parenting with self-esteem. In other words, whether authoritative parenting or positive family relations show stronger associations with child outcomes may depend on the kind of assessment on parenting style and family relations. This point should at least be mentioned in the Limitations section.
In addition, I have some minor comments:
At page 4 (line 151) the authors wrote that externalizing behaviors develop from the overall family socialization experiences. However, this view is much too narrow as personality characteristics of the young people as well as social experiences outside the family also contribute to externalizing problem behavior.
Please provide more convincing arguments for the suggestion that family processes should be more influential on externalizing problems than authoritative parenting. As authoritative parenting includes high levels of monitoring and parental attempts to prevent or reduce problematic behaviors, parenting styles could be expected to have stronger effects on externalizing problems than more general family processes. The only exception might be if the family processes focus mainly on aggression within the family.
Please specify what current status means (p. 4, line 187).
At page 4 (line 191) the authors write that female youths tend to have more behavioral problems than male youths. However, because the authors focus on externalizing problems, it would be better to cite research showing that male young people show higher levels of externalizing problems than their female peers.
In the footnote of page 5, the authors wrote that their recruited Chinese parent-use dyads via local churches from Hong Kong. However, this is very unlikely to lead to a representative sample.
More information on the meaning of the variable “future orientation” would be needed. The example refers to self-control rather than to future orientation in general.
Authoritative parenting and positive family processes were correlated at r=.72. This indicates that both variables share about 50% of their variance, thus indicating multi-collinearity. It would be more informative if the authors had assessed two variables that did not overlap so much.
In Table 4, the term “effective parenting” should probably read “authoritative parenting”.
I do not think that the present study sheds light on existent family intervention programs (as the authors stated at page 10 and 11, line 384 to 385). The authors did not assess any intervention program. Please be more specific with regard to your conclusions.
All references have to be checked. For example the study by Tabia and Marzocchi on the Italian version of the SDQ does not describe the Parental Authority Questionnaire by Buri (1991). Reference Nr 28 does not refer to the Positive Self-Image Scale (as stated by the authors).
In sum, I cannot recommend the manuscript in its present form for publication. A revised manuscript could maintain the analyses on associations of family relations and authoritative parenting with youth outcomes. However, better statistical tests for differences in regression coefficients would be needed. I would suggest deleting the mediator analysis. Alternatively, the authors would have to add analyses with other mediators and check whether the different analyses show similar model fit or whether one model fits the data best. In addition, the authors would have to avoid any causal interpretations of their correlational data. The suggested changes could lead to a manuscript that might be suitable for a publication in the International Journal of Environmental Research and Public Health.
Author Response
After examining the written comments of reviewer 2, we have revised the manuscript point by point and our replies are following
1) First, I did not find the hypothesis about moderating effects very convincing. In addition, concurrent correlational data do not provide a stringent test for mediation. Many alternative mediation models might fit the data as well. For example, authoritative parenting might promote self-control, which again reduces the level of externalizing problem behavior. In other words, the authors test one of a number of possible mediation models and it is not clear whether the tested model fits the data best.
Reply: the present study does not intend to do moderating tests on examining whether positive family processes would moderate effects of authoritative parenting on youth outcomes, or the reverse. In fact, one of the aims of this study is to compare whether positive family processes and authoritative parenting would have significant different effects on respective youth outcomes. For this, we have rephrased the presentations to make the comparative purpose clearer, in which we wrote
“As authoritative parenting and family processes are two different facets of parental socialization, research should investigate their respective effects on youth development. Hence, in this study it is expected that positive family processes would predict youths’ various developmental outcomes.”(lines 115-118)
In addition, for other possible mediations of youth outcomes, such as the mediating effects of youth self-control on the relationships between parental socialization and other youth outcomes, we do not exclude this possibility. Nevertheless, the current study mainly focuses on the mediating effects of youth self-esteem on their developmental outcomes, which, we consider, are plausible and justifiable theoretically and practically. This is because, as mentioned in the manuscript, self-esteem is an intrapersonal construct of cognitive and psychosocial nature that is profoundly influential of youths’ other aspects of development, which include their self-control, future orientation, other perspective taking and externalizing problem behavior. Therefore, the mediating tests conducted in this study are not necessary in contradiction to the reviewer’s proposition regarding other alternative mediations between parental socialization and youth outcomes. For this, we added additional arguments to fortify testing the mediating effects of youth self-esteem, which is
“Generally, self-esteem is an important cognitive and psychosocial construct that may sway how individuals evaluate themselves, judge others and interpret external social events around them, which cohere with the perspective of self-referent cognitions suggesting that youths’ emotional expressions, psychological responses, behavioral reactions, apprehension of external environment and future aspiration are all essentially contingent on their referring to self-identity, worth and competence [19]. Albeit other youths’ cognitive and psychological traits, such as youth self-control, holding the possibility to mediate the relationships between parental socialization and youth development, youth self-esteem is considered as a crucial mediator, which is because self-image and concept are found more stable and influential on other cognitive, psychosocial, and behavioral choices and responses, leading to an individual’s overall development and consequences [11,17]. Thus, if youths are of higher self-esteem, they would develop better self-control, future orientation, perspective taking behavior, and lower externalizing problems [20,21]. Furthermore, as family is a fundamental nurturing context to develop youth self-esteem mainly through parental socialization [1,17]; hence, it is believed that youth self-esteem would mediate in the relationships between parental socialization and youth outcomes.” (Lines 136-152)
In fact, we have conducted other alternative mediational tests, as suggested by the reviewer, and found that although youth self-control has significant mediating effects on the relationships between parental socialization and youth outcomes of self-esteem, future orientation, other perspective taking and externalizing problem behavior (Table 1), other youth outcomes do not show all these mediating effects (Table 2). Hence, as youth self-esteem of theoretical importance for youth development, we decide to keep the indirect effects of youth self-esteem in the relationships between parental socialization and youth outcomes (note that modeling indirect/ mediating effects is not conducted by separate mediating tests, but pooling the exogenous predictor of authoritative parenting/ positive family process, the mediator of youth self-esteem and the youth outcomes in one single modeling procedure by a multivariate approach).
Table 1. Mediating effects of youth self-control on the relationships between parental socialization and youth outcomes | |||||
Authoritative parenting | Indirect effect (βind) | SE | z-value | Effect size (k2) | |
1. | Self-Esteem | .181** | .064 | 2.831 | .152 |
2. | Future Orientation | .208** | .073 | 2.871 | .187 |
3. | Other Perspective Taking | .192*** | .064 | 2.983 | .144 |
4. | Externalizing Problem Behavior | -.119* | .044 | -2.673 | .087 |
Positive family processes | Indirect effect (βind) | SE | z-value | Effect size (k2) | |
1. | Self-Esteem | .182** | .066 | 2.737 | .140 |
2. | Future Orientation, | .196** | .072 | 2.733 | .170 |
3. | Other Perspective Taking | .168** | .062 | 2.726 | .130 |
4. | Externalizing Problem Behavior | -.113* | .045 | -2.502 | .088 |
*p< .05; **p< .01; ***p< .001 | |||||
Table 2. Mediating effects of youth Other perspective taking on the relationships between parental socialization and youth outcomes | |||||
Authoritative parenting | Indirect effect (βind) | SE | z-value | Effect size (k2) | |
1. | Self-Esteem | .127*** | .031 | 4.047 | .140 |
2. | Self-control | .136*** | .032 | 4.252 | .156 |
3. | Future Orientation | .140*** | .032 | 4.333 | .156 |
4. | Externalizing Problem Behavior | -.019 | .023 | -.824 | .039 |
Positive family processes | Indirect effect (βind) | SE | z-value | Effect size (k2) | |
1. | Self-Esteem | .152* | .060 | 2.529 | .117 |
2. | Self-control | .157* | .061 | 2.563 | .129 |
3. | Future Orientation | .149* | .059 | 2.526 | .128 |
4. | Externalizing Problem Behavior | -.061 | .036 | -1.712 | .032 |
*p< .05; **p< .01; ***p< .001 | |||||
2) Second, the statistical test for differences in regression coefficients is not adequate. The authors should provide the 95% confidence intervals of the regression coefficients. If the intervals overlap, this indicates that the associations of authoritative parenting and of positive family relations with the child outcome do not differ significantly. I would guess that this will be the case in most or even all tests.
Reply: As mentioned above, this study employed multivariate regression modeling to regress all the youth outcomes in a single modeling procedure, which can adjust bias of type 1 errors. Nevertheless, we now provide the regression results with 95% confidence intervals below, in which the significant results correspond to the regression tables that report t-values. In fact, we have now substituted the 95% CI regression tables for the t-value regression tables.
Table 3. Multivariate linear regression modeling predicting self-concept, self-control, future orientation, other perspective taking and externalizing problem behaviors of youth children by authoritative parenting | ||||||||||
Self-esteem | Self-control | Future orientation | Other perspective taking | Externalizing problem behavior | ||||||
β | 95% CI | β | 95% CI | β | 95% CI | β | 95% CI | β | 95% CI | |
Non-intact family | -.126 | -.274, .022 | -.105 | -.251, .040 | -.092 | -.234, .052 | .003 | -.142, .147 | .022 | -.120, .164 |
Welfare dependence | .087 | -.050, .223 | .142* | .008, .276 | .011 | -.121 .143 | .065 | -.068, .198 | -.052 | -.183, .079 |
Family members | -.093 | -.234, .046 | -.027 | -.165, .109 | -.102 | -.237, .033 | .046 | -.090, .183 | .062 | -.072, .196 |
Child gender | .019 | -.110, .147 | .017 | -.109, .143 | .110 | -.014, .234 | .025 | -.100, .150 | -.196** | -.318, -.072 |
Child age | -.048 | -.286, .190 | -.160 | -.394, .073 | -.236* | -.466, -.006 | -.072 | -.302, .161 | .122 | -.107, .349 |
Child status | .115* | -.239, .010 | -.072 | -.195, .050 | -.052 | -.172, .069
| -.103* | -.224, .019 | .274* | .154, .394 |
Child education | .137 | -.102, .376 | .244* | -.009, .478 | .335** | .104, .566 | .207+ | -.026, .440 | -.128 | -.357, .102 |
Authoritative parenting | .258*** | .127, .389 | .312*** | .184, .440 | .273*** | .147, .400 | .346*** | .218, .473 | -.206** | -.331, -.080 |
R2 | .111 | .144 | .170 | .159 | .183 | |||||
Model X2(df) | 377.729(50) *** | |||||||||
*p< .05; **p< .01; ***p< .001. | ||||||||||
Table 4. Multivariate linear regression modeling predicting self-concept, self-control, future orientation, other perspective taking and externalizing problem behaviors of youth children by positive family processes. | ||||||||||
Self-Esteem | Self-control | Future orientation | Other perspective taking | Externalizing problem behavior | ||||||
β | 95% CI | β | 95% CI | β | 95% CI | β | 95% CI | β | 95% CI | |
Non-intact family | -.098 | -.245, .047 | -.096 | -.243, .051 | -.089 | -.243, .058 | .007 | -.141, .155 | .006 | -.136, .148 |
Welfare dependence | .070 | -.063, .201 | .112 | -.022, .245 | -.018 | -.150, .115 | .029 | -.105, .163 | -.036 | -.165, .092 |
Family members | -.092 | -.229, .044 | -.040 | -.178, .097 | -.117 | -.253, .020 | .028 | -.110, .167 | .065 | -.068, .197 |
Child gender | .037 | -.088, .162 | .040 | -.086, .167 | .131* | .006, .256 | .051 | -.076, .178 | -.211** | -.332, -.088 |
Child age | -.081 | -.311, .149 | -.215* | -.448, .017 | -.287* | -.518, -.057 | -.136 | -.369, .098 | .151 | -.073, .375 |
Child status | -.102+ | -.223, .202 | -.061 | -.184, .063 | -.043 | -.165, .079 | -.092* | -.215, .033 | .265* | .146, .383 |
Child education | .144 | -.088, .376 | .275* | .040, .510 | .368** | .135, .600 | .249* | .013, .484 | -.139 | -.365, .087 |
Positive family processes | .326*** | .201, .451 | .282*** | .156, .409 | .220*** | .095, .345 | .279*** | .153, .407 | -.233*** | -.355, -.111 |
R2 | .151 | .131 | .148 | .125 | .196 | |||||
Model X2(df) | 377.926(50)*** | |||||||||
*p< .05; **p< .01; ***p< .001 | ||||||||||
Table 5. Multivariate linear regression modeling predicting self-control, future orientation, other perspective taking and externalizing problem behaviors of youth children by their self-concept and authoritative parenting. | ||||||||
Self-control | Future orientation | Other perspective taking | Externalizing problem behavior | |||||
β | 95% CI | β | 95% CI | β | 95% CI | β | 95% CI | |
Non-intact family | -.052 | -.183, .050 | -.060 | -.199, .080 | .047 | -.089, .182 | .001 | -.140, .141 |
Welfare dependence | .105* | -.016, .226 | -.011 | -.138, .117 | .035 | -.089, .160 | -.037 | -.167, .092 |
Family members | .102 | -.112, .136 | -.079 | -.210, .053 | .079 | -.049, .207 | .046 | -.087, .179 |
Child gender | .009 | -.105, .122 | .105 | -.015, .225 | .018 | -.098, .135 | -.193** | -.313, -.071 |
Child age | -.140 | -.350, .070 | -.224* | -.445, -.001 | -.055 | -.270, .162 | .114 | -.112, .337 |
Child status | -.023 | -.134, .088 | -.023 | -.140, .094 | -.063 | -.177, .052 | .255*** | .136, .373 |
Child education | .185+ | -.027, .397 | .300* | .076, .524 | .160 | -.059, .377 | -.105 | -.330, .122 |
Self-esteem | .428*** | .312, .544 | .251*** | .129, .374 | .348*** | .228, .467 | -.170** | -.294, -.046 |
Effective parenting | .202** | .082, .321 | .208** | .082, .335 | .256*** | .133, .379 | -.162* | -.289, -.034 |
R2 | .307 | .226 | .266 | .209 | ||||
Model X2(df) | 345.558(42)*** | |||||||
*p< .05; **p< .01; ***p< .001 | ||||||||
Table 6. Multivariate linear regression modeling predicting self-control, future orientation, other perspective taking and externalizing problem behaviors of youth children by their self-concept and positive family processes. | ||||||||
Self-control | Future orientation | Other perspective taking | Externalizing problem behavior | |||||
β | 95% CI | β | 95% CI | β | 95% CI | β | 95% CI | |
Non-intact family | -.053 | -.187, .081 | -.063 | -.204, .079 | .042 | -.097, .181 | -.009 | -.149, .132 |
Welfare dependence | .082 | -.039, .203 | -.036 | -.163, .093 | .004 | -.121, .130 | -.026 | -.153, .101 |
Family members | .000 | -.126, .125 | -.093 | -.225, .040 | .061 | -.068, .192 | .051 | -.081, .182 |
Child gender | .024 | -.090, .139 | .121 | .000, .242 | .038 | -.081, .157 | -.205 | -.325, -.084 |
Child age | -.180 | -.391, .030 | -.267 | -.489, -.043 | -.107 | -.325, .112 | .139 | -.083, .360 |
Child status | -.017 | -.129, .095 | -.017 | -.136, .102 | -.055 | -.171, .062 | .249* | .131, .367 |
Child education | .213* | -.001, .426 | .330** | .105, .556 | .197+ | -.025, .418 | -.117 | -.341, .107 |
Self-esteem | .432*** | .312, .552 | .259*** | .131, .385 | .358*** | .233, .482 | -.150* | -.276, -.024 |
Positive family processes | .142* | .021, .262 | .135* | .008, .264 | .163** | .038, .289 | -.184** | -.311, -.057 |
R2 | .290 | .205 | .233 | .215 | ||||
Model X2(df) | 336.071(42)*** | |||||||
*p< .05; **p< .01; ***p< .001 | ||||||||
3) Third, the authors provide causal interpretations of correlational data which is highly inadequate. Correlations cannot be interpreted as indicating effects or influences of authoritative parenting or positive family relationship on the child outcome as there are also child effects on the quality of family relations and on parenting styles. The authors cite one meta-analysis that has shown that there are bi-directional effects between parenting and parent-child relationship and child outcomes.
Reply: we agree with the reviewer that cross-sectional are inadequate in explaining the casual relationships between parental socialization and youth development, and there were also some scholars assume the bi-directional relationships between family socialization and youth behaviors. However, we have apparently reported in the discussion of our study that due to the empirical results generated from cross-sectional data, which “makes causal validity impossible (line 517)”, and, in fact, we have attempted to avoid any interpretations in the manuscript to project temporal causality for the relationships between parental socialization and youth development.
In addition, regarding the reciprocal relationships between parental socialization and youth psychological and behavioral development, which are manifestly less conclusive than the direct impacts of parental socialization on youth development. Although there are some pertinent studies that found the bi-directional parent-child relationships (Burke, Pardini, & Loeber, 2008; Bell & Belsky, 2008; Serbin, Kingdon, Ruttle, Stack, 2015; Williams & Steinberg, 2011), other research did not concretely support these bi-directional relationships (Gouze, Hopkins, Bryant, & Lavigne, 2017; Rolon-Arroyo, Arnold, Breaux, & Harvey, 2018; Stepp et al., 2014). Nevertheless, compared to parenting effects on child development, effects of child psychological and behavioral development on parental socialization are found much weaker, albeit significant (Gouze et al., 2017; Williams & Steinberg, 2011). Furthermore, even significant bi-directional relationships found between parental socialization and child psychological and behavioral development, these significant findings are mixed, in which some research found that youth and children’s maladjustment might undermine positive parenting practices (Bell & Belsky, 2008; Gouze et al., 2017), while other research reported that higher child maladjustment would lead to better parental socialization, which in turn alleviates problematic development of their offspring (Serbin et al., 2015). Thereby, we think it is plausible to investigate effects of parenting practices and family processes on youth development in Chinese context, as the current study did. This is because parental socialization has been universally accepted as the most important cultivating precursor to shape cognitive, psychological and behavioral outcomes of children and youths, which happens even at the very early life stages of children before they are capable of confronting their parents regarding their parenting practices. As such, despite the fact that youths’ psychological and behavioral outcomes would affect parenting practices, it is not necessary to contradict researching effects of parental socialization on youth development. For this, we would address the reviewer’s point in the Discussion part to remind readers the possibility of bi-directional relationships existing in parental socialization and youth development, which include
“Furthermore, this study only examined effects of authoritative parenting and positive family processes on youth outcomes, and the possibility of reciprocal relationships between parental socialization and youth development has not yet explored [33,34]. Future research should consider how parents and youths mutually affect and reinforce each other in a dynamic way to enhance current knowledge on parental socialization.” (Lines 520-524).
Bell, B. G., & Belsky, J. (2008). Parents, parenting, and children’s sleep problems: Exploring reciprocal effects. British Journal of Developmental Psychology, 26, 9-593.
Burke, J. D., Pardini, D. A., & Loeber, R. (2008). Reciprocal Relationships Between Parenting Behavior and Disruptive Psychopathology from Childhood Through Adolescence. Journal of Abnormal Child Psychology, 36, 679-692.
Gouze, K. R., Hopkins, J., Bryant, F. B., & Lavigne, J. V. (2017). Parenting and Anxiety: Bi-directional Relations in Young Children. Journal of Abnormal Child Psychology, 45, 1169-1180.
Rolon-Arroyo, B., Arnold, D. H., Breaux, R. P., Harvey, E. A. (2018). Reciprocal Relations Between Parenting Behaviors and Conduct Disorder Symptoms in Preschool Children. Child Psychiatry & Human Development, 49, 786-799.
Serbin, L. A., Kingdon, D., Ruttle, P. L., Stack, D. M., (2015). The impact of children's internalizing and externalizing problems on parenting: Transactional processes and reciprocal change over time. Development and Psychopathology, 27, 969-986.
Stepp, S. D., Whalen, D. J., Scott, L. N., Zalewski, M., Loeber, R., & Hipwell, A. E. (2014). Reciprocal effects of parenting and borderline personality disorder symptoms in adolescent girls. Development and Psychopathology, 26, 361-378.
Williams, L. R., & Steinberg, L. (2011). Reciprocal Relations Between Parenting and Adjustment in a Sample of Juvenile Offenders. Child Development, 82, 633-645.
4) Fourth, the authors use a less than optimal measure of authoritative parenting (the Parental Authority Questionnaire). I am aware of the fact that this measure is often used. However, Buri’s scale mainly assesses only control while the conceptualization of authoritative parenting by Maccoby and Martin conceptualize this parenting style by a combination of control and parental warmth. If the authors had used a measure of authoritative parenting that also includes parental warmth, they might have found stronger associations of authoritative parenting with self-esteem. In other words, whether authoritative parenting or positive family relations show stronger associations with child outcomes may depend on the kind of assessment on parenting style and family relations. This point should at least be mentioned in the Limitations section.
Reply: Agree, we now added statements of the shortcomings of using Authoritative Parenting subscale of the Parental Authority Questionnaire (PAQ) to measure authoritative parenting in the limitations section of the Discussion part, in which we wrote
“In addition, it should be noted that validity of the results would be affected by the measurement instruments and scales adopted to measure the study variables. In this study, authoritative parenting was measured by Authoritative Parenting subscale of the Parental Authority Questionnaire that is more emphasized on measuring parental control rather than a combination of parental demandingness and support. It is suggested to employ more well-established measures in the future to tap on authoritative parenting, such as Parenting Styles and Dimensions Questionnaire (PSDQ) [35].” (Lines 524-530)
Minor revisions
5) At page 4 (line 151) the authors wrote that externalizing behaviors develop from the overall family socialization experiences. However, this view is much too narrow as personality characteristics of the young people as well as social experiences outside the family also contribute to externalizing problem behavior. Please provide more convincing arguments for the suggestion that family processes should be more influential on externalizing problems than authoritative parenting. As authoritative parenting includes high levels of monitoring and parental attempts to prevent or reduce problematic behaviors, parenting styles could be expected to have stronger effects on externalizing problems than more general family processes. The only exception might be if the family processes focus mainly on aggression within the family.
Reply: agree, we have now revised the contents presented in line 151 of page 4 to make the meanings more prominent regarding to the stronger effect of positive family processes on youth externalizing problem behavior than that of authoritative parenting, in which we now wrote
“Therefore, it is plausible that these two facets of parental socialization may contribute to youth development differentially. Specifically, youth self-esteem is a cognitive and psychosocial construct that is apparently cultivated and shaped through constant social experiences, interpersonal interactions, and establishment of human trust and worth within the family realm [1,20]. Moreover, family relationships and home climate provide one of the most robust modeling environments influential on youths’ behavioral patterns and choices [2,7,9], especially for out-of-home behavioral decisions and manners, which need individual sagacity and judgment to assess and select correct responses [3,7], e.g. deciding to engage in deviance or not. Thereby, incessant and prevailing experiences of family processes are expected more influential on youths’ self-esteem and externalizing problem behavior than that of authoritative parenting that appears acting out more conditionally and situationally.”(Lines 170-180)
6) At page 4 (line 191) the authors write that female youths tend to have more behavioral problems than male youths. However, because the authors focus on externalizing problems, it would be better to cite research showing that male young people show higher levels of externalizing problems than their female peers.
Reply: our writings from line 190 to 192 are that “Additionally, empirical literature pointed out that female youths tend to have higher psychological difficulties and their male counterparts are apt to have more behavioral problems but higher self-esteem [2,16].”(now is lines 222-224) Hence, we intend to mention that male youths would have higher behavioral problems than their female counterparts; and now we cited more research to support this saying.
7) In the footnote of page 5, the authors wrote that their recruited Chinese parent-use dyads via local churches from Hong Kong. However, this is very unlikely to lead to a representative sample.
Reply: we did not mention the sample that is a representative one in the manuscript, and now we added a sentence to connote that the recruited sample is a non-random community sample in the Discussion part, in which we wrote
“Third, data for analysis in this study was obtained from a nonrandom sample of Chinese parent-youth dyads recruited through local churches that may restrict variances of the study variables and representativeness of the results.”(Lines 517-520)
8) More information on the meaning of the variable “future orientation” would be needed. The example refers to self-control rather than to future orientation in general.
Reply: more information about the variable of “future orientation” has been added to make it more understandable as measuring youths’ future expectation and orientation, in which we wrote
“Youth future orientation was measured by the 8-item Future Outlook Inventory [30], which is to measure future expectations and aspiration (Cauffman & Steinberg, 2000). An example items include “I think about how things might be in the future” and “I make lists of things to do”. Internal reliability in this study was highly satisfactory, α= .78.”
9) Authoritative parenting and positive family processes were correlated at r=.72. This indicates that both variables share about 50% of their variance, thus indicating multi-collinearity. It would be more informative if the authors had assessed two variables that did not overlap so much.
Reply: we now have added evidence in the Results part to prove the variables of authoritative parenting and positive family processes that are not overlapped, in which we include new evidence to show that authoritative parenting and positive family processes are two discrete but mutually related family constructs, in which we wrote
“Notably, authoritative parenting was strongly correlated with positive family processes, r= .719, p< .001, which is within the acceptable correlational threshold, that’s r≤ .80, for preclusion of concern of collinearity. For further proving authoritative parenting and positive family processes that are two discrete but mutually related family constructs, both confirmatory factor analysis with item parcels and multiple indicators multiple causes (MIMIC) modeling were applied. First, due to the scales used to measure authoritative parenting and positive family processes having many items, Authoritative Parenting subscale of the Parental Authority Questionnaire was randomly parceled into three indicators starting by a seed number of 3, and Family Functioning Style Scale was parceled into four indicators by a seed number of 4, in which the latent factors of authoritative parenting and positive family processes were allowed to be correlated and MIMIC modeling permitted regressing covariates of family composition, family welfare dependence, number of family members, youth gender, age, current status and educational levels on the factor model. As a result, the CFA model obtained an excellent model-data fit, CFI= .998, TLI=.997, RMSEA= .028, X2= 15.309, df=13, p=.288, with factor loadings ranged from λ=.829 to .895 for authoritative parenting and factor loadings ranged from λ=.884 to .923 for positive family processes. In addition, the MIMIC model also obtained a very good model-data fit, CFI= .994, TLI=.991, RMSEA= .029, X2= 57.101, df=48, p=.172, with factor loadings ranged from λ=.826 to .900 for authoritative parenting and factor loadings ranged from λ=.885 to .921 for positive family processes.” (Lines 332-349)
10) In Table 4, the term “effective parenting” should probably read “authoritative parenting”.
Reply: we have now rephrased the term “effective parenting” to “authoritative parenting”.
11) I do not think that the present study sheds light on existent family intervention programs (as the authors stated at page 10 and 11, line 384 to 385). The authors did not assess any intervention program. Please be more specific with regard to your conclusions.
Reply: we now rewrite the presentations to appeal the needs to note the importance of enhancing family processes rather than emphasizing on improving parenting only, in which we wrote
“As a result, parental socialization should not only emphasize on parenting practices solely but also consider other important facts of parental socialization in contribution to positive youth development, such as positive family processes.” (Lines 471-474).
12) All references have to be checked. For example the study by Tabia and Marzocchi on the Italian version of the SDQ does not describe the Parental Authority Questionnaire by Buri (1991). Reference Nr 28 does not refer to the Positive Self-Image Scale (as stated by the authors).
Reply: we now have checked the references to ensure their correctness, and for Parental authority questionnaire, for authoritative parenting we cited
[25] Buri, J.R. Parental authority questionnaire. Journal of Personality Assessment 1991, 57, 110-119.
For youth positive self-image scale, we cited (which provided also the scale items)
[28] Regnerus, M.D.; Elder, G.H. Staying on Track in School: Religious Influences in High- and Low-Risk Settings. Journal for the Scientific Study of Religion 2003, 42, 633-649.

Round 2
Reviewer 1 Report
the discussion and the results have improved it has been improved in the limitations
Author Response
After taking comments from the reviewer 1, we have again revised the current manuscript with the following amendments:
The whole manuscript has been carefully read again and some presentations are modified to ensure its accuracy and precisions in conveying logical arguments.
Results of the study has been checked again to ensure the correctness of the findings.
All spellings are re-checked to confirm their corrections
The Discussion part has modified, especially for the limitations section in it, to make implications and insights of the findings of the current study more comprehensive and thoughtful.
Reviewer 2 Report
The revised manuscript still contains an incorrect test of significance of the differences between associations of parenting and family processes with child outcomes. The table incorrectly reports that the size of associations of parenting would differ from the size of association of family processes. The standard errors of the individual regression coefficients are larger than the size of the differences between two regression coefficients. The 95%-confidence intervals of the regression coefficients also overlap -- both statistics indicate that the size of the Regression coefficient does not differ significantly. Thus, I strongly recommend deleting Table 7 and not claiming that these coefficients differ.
I also still suggest that the authors should add to their Limitations section that they assessed only one possible mediator - self-esteem. Alternative (untested) mediator models may show a similar model fit, such as authoritative parenting promoting self-control which again reduces externalizing problem behavior..
Author Response
For reviewer 2, we have made the following amendments:
English presentations of the whole manuscript have been proofread again in order to make sure precision and authenticity of the sentence and grammatical structures.
Results of all multivariate regression models have been carefully checked again and some modifications has been made in the tables (Table 4 & 6) to correct the typos.
All regression coefficients and 95% CIs in the multivariate regression models have been checked and confirmed correctly, and all the multivariate regression models are re-run again to ensure that their results are in alignment with the previous findings.
We earnestly request to keep Table 7 for the purpose of comparing significantly different effects of authoritative parenting and positive family processes on youth outcomes, which, we thought, would contribute to the literature of family research when comparing influences of different dimensions of parental socialization on youth development. For this, we added a new sentence to summarize the different influences of authritative parenting and positive family processes on various youth outcomes differently:
“In sum, although both authoritative parenting practices and positive family processes were found significantly predictive of various aspects of youth development, their different effects on different youth outcomes were verified.” [Lines 421 to 424]
Now, we put new discussion of other possible “Alternative (untested) mediator models” in the limitations section of the Discussion part with writings of
“In addition, the current study only examined mediating effects of youth self-esteem on the relationships between parental socialization and youth development, and other possible mediators are yet to be explored. For example, there is a possibility of youth self-control in mediation of effects of parental socialization on youth externalizing behaviors [3,5]. Therefore, future research should consider to adopt a multiple-mediation model to incorporate other possible and influential mediators for the associations between parental socialization and youth development. Lastly, parental socialization only refers to authoritative parenting and positive family processes, and other possible facets of family socialization experiences have not included in this study, e.g. other parenting styles and parent-youth disclosure, would circumscribe our understandings of how parental socialization affects youth development more comprehensively. If future research can address these limitations, a more dynamic and exhaustive picture of parental socialization and youth development can attain.”[Lines 493-504]